# *In-Vitro* and *In-Vivo* Antibacterial Effects of Frankincense Oil and Its Interaction with Some Antibiotics against Multidrug-Resistant Pathogens

**DOI:** 10.3390/antibiotics11111591

**Published:** 2022-11-10

**Authors:** Megren Bin Faisal Almutairi, Mohammed Alrouji, Yasir Almuhanna, Mohammed Asad, Babu Joseph

**Affiliations:** 1Department of Clinical Laboratory Science, College of Applied Medical Sciences, Shaqra University, Shaqra 11961, Saudi Arabia; 2Department of Clinical Laboratory, King Saud Hospital, Unaizah 56215, Saudi Arabia

**Keywords:** pneumonia, volatile oils, *Boswellia*, MIC, MBC

## Abstract

Frankincense (*Boswellia sacra* oleo gum resin) is reported to possess antimicrobial activity against several pathogens in-vitro. The antimicrobial effects of frankincense oil and its interaction with imipenem and gentamicin against methicillin-resistant *Staphylococcus aureus* (MRSA) and multidrug-resistant *P. aeruginosa* were determined through in-vitro methods and an in-vivo study using a rat pneumonia model. Frankincense oil was subjected to GC-MS analysis to determine the different volatile components. Antibacterial effects against MRSA and MDR-*P. aeruginosa* was evaluated and its MIC and MBC were determined. For the rat pneumonia model (in-vivo), oil was administered at a dose of 500 mg/kg and 1000 mg/kg followed by determination of CFU in lung tissue and histological studies. Frankincense oil did not show a very potent inhibitory effect against MRSA or MDR-*P. aeruginosa*; the oil did not affect the zone of inhibition or FIC when combined with imipenem or gentamicin indicating a lack of interaction between the oil and the antibiotics. Furthermore, there was no interaction between the antibiotics and the frankincense oil in the in-vivo model. The result of the study revealed that frankincense oil has a weak inhibitory effect against MRSA and MDR-*P. aeruginosa*, and it did not show any interaction with imipenem or gentamicin.

## 1. Introduction

Antimicrobial resistance (AMR) is a threat to public health that leads to the development of serious infections and interferes with strategies for the prevention of infections [1]. The AMR is continuously increasing and is showing no signs of slowing down. Factors such as spontaneous evolution, mutation, and transfer of resistant genes contribute significantly to AMR [2,3].

Herbal medicines and their constituents have been increasingly evaluated to combat antimicrobial resistance and to explore new and more potent antimicrobial agents with lesser adverse effects. Several plant-derived components have exhibited remarkable inhibitory properties on microbial growth [4,5,6]. Frankincense (*Boswellia sacra* oleo gum resin) is reported to possess antimicrobial activity in earlier studies. Oil extracted from frankincense grown in different regions of Oman was reported to inhibit *S. aureus* and *P. aeruginosa* in-vitro [7]. Frankincense oil from 20 different countries showed varying effects on 5 different micro-organisms; *S. aureus*, *B. cereus*, *E. coli*, *P. vulgaris*, and *C. albicans* [8]. An earlier study about the interaction of frankincense oil with myrrh oil showed that frankincense oil produces a synergistic or additive effect with myrrh oil depending on the type of micro-organism [9]. All of these studies were carried out using in-vitro methods and no attempt has been made so far to evaluate the antibacterial effect of frankincense oil in-vivo after oral administration.

Methicillin-resistant *Staphylococcus aureus* (MRSA) causes both nosocomial and community-acquired infections. Several clones of MRSA have emerged recently and its resistance to previously sensitive antibiotics is a matter of concern for healthcare community workers [10,11]. Multidrug-resistant *Pseudomonas aeruginosa* (MDR-*P. aeruginosa*) causes severe infections and has an outstanding ability for being selected and for spreading antimicrobial resistance in-vivo [12,13]. Furthermore, the spread of “high-risk” clones of MDR-*P. aeruginosa* is a threat to global public health [14].

The present study determined the antimicrobial effect of frankincense oil and its interaction with imipenem and gentamicin against MRSA and multidrug-resistant *P. aeruginosa* using in-vitro methods and an in-vivo study in a rat pneumonia model. Further, the study also evaluated its interaction with commonly used antibacterial agents. Lastly, the pathogens that were resistant to many commonly used antibacterial agents were employed in this study.

## 2. Results

### 2.1. GC-MS Analysis of Frankincense Oil

The analysis revealed the presence of 40 constituents (Table 1). Compounds known to be present such as α-pinene, camphene, and limonene were present in the oil. The maximum area under the curve was observed for α-pinene and the minimum area under the curve was seen with β-myrcene indicating the most prominent and least amount of volatile components in the oil, respectively (Figure 1).

### 2.2. Antibacterial Effect of Frankincense Oil

Frankincense oil did not show a very potent inhibitory effect against MRSA or MDR-*P. aeruginosa* and a noticeable zone of inhibition could be seen starting from 20 µL. Furthermore, the oil did not affect the zone of inhibition induced by imipenem or gentamicin, indicating a lack of interaction between the oil and the antibiotics (Figure 2A).

Values are mean ± SEM for three independent trials. Antibiotic refers to imipenen (4 µg/mL) and gentamicin (10 µg/mL) against MRSA and MDR-*P. aeruginosa*, respectively.

The MIC of frankincense oil was 10 µL and 20 µL against MRSA MDR-*P. aeruginosa*, respectively. The MBC was found to be 20 µL for MRSA and 40 µL for MDR-*P. aeruginosa* (Figure 2B).

### 2.3. Interaction Study of Frankincense Oil with Antibiotics against MDR Strains

The FIC determined to evaluate the interaction between the frankincense oil and the antibiotics showed no interaction between the oil and imipenem or gentamicin (Table 2).

### 2.4. Rat Pneumonia Model (In-Vivo)

The frankincense oil at both doses significantly reduced the bacterial load of MRSA in the rats’ lungs after treatment for 4 days. However, it was less effective than the standard antibiotics; gentamicin and imipenem. In rats infected with MDR-*P. aeruginosa*, the lower dose of the oil (500 mg/kg) was ineffective and the effect of the higher dose of frankincense oil (1000 mg/kg) was relatively less effective compared to its effect against MRSA infection. Furthermore, there was no interaction between the antibiotics and the frankincense oil in the in-vivo model and this was similar to the results obtained in-vitro (Figure 3). Since the duration of the experiment was only 4 days, all the animals survived during the experimental period.

Histological examination of tissues from rats of different groups showed that both MRSA and MDR-*P. aeruginosa* infection had caused lung inflammation and the lung tissue was infiltrated with fluids and inflammatory cells. This inflammatory response was suppressed by frankincense and standard antibiotics. The histological changes are shown in Figure 4a–g. There was no noticeable change in the histology of lung tissue after treatment with antibiotics and frankincense oil when compared to individual treatments. Hence, photomicrographs showing the effect of combination treatment are not shown.

## 3. Discussion

The antibacterial effects of frankincense oil have been reported earlier against a variety of pathogens [7,8]. However, this study is different from the earlier reports in many aspects. Frankincense oil has been used for an antibacterial effect by the conventional in-vitro methods, the present study determined its antibacterial effect after oral administration (in-vivo) along with in-vitro studies.

Frankincense oil showed modest antibacterial activity against MRSA and MDR-*P. aeruginosa* but there was no interaction between the oil and imipenem or gentamicin in both in-vitro and in-vivo studies. The pathogens causing common infections and resistant to many of the commonly used antibacterial agents were selected for the study.

The frankincense is given different names, such as *Boswellia serrata* for Asian and African frankincense and *Boswellia sacra* for frankincense obtained from Oman, a Middle Eastern country. In our case, we selected oil that was prepared from *Boswellia sacra* for our study. We analyzed the chemical constituents present in the commercially obtained oil by GC-MS so that if a study is repeated with frankincense having the same constituents, a similar effect may be obtained. Furthermore, analysis of the oil revealed the probable constituent(s) that might have contributed to the effect.

The in-vivo evaluation of herbal drugs is important because herbs are administered only by oral route due to several different chemical constituents present in them [15]. The oil was administered as such in the study to mimic its traditional use [16], and no derivatization was done to make it more polar for oral administration. It is well known that many antibacterial agents are not effective orally either because they are not absorbed or some of them undergo extensive first-pass metabolism or get destroyed by acid/enzymes in the digestive tract [17]. In the current study, the antibacterial effect of frankincense oil observed in-vivo after oral administration was similar to that observed in-vitro, though the effect was not very potent in both studies. This suggests that active constituent(s) of frankincense oil responsible for the antibacterial effect are well absorbed orally and reach the blood circulation in sufficient amounts to exert their effect. Frankincense oil is used traditionally in the treatment of respiratory diseases through oral administration and is considered safe for oral consumption [16]. We selected the doses of frankincense oil based on earlier reports [18]. No behavioral difference or toxicity was observed between the different treated groups.

As mentioned above, the essential oils of frankincense have been reported for antibacterial effects against many pathogens earlier. However, many of the earlier reports determined the antibacterial effects with common bacteria, and no attempt was made to evaluate the effect of essential oils from frankincense on MRSA and MDR-*P. aeruginosa*. It was shown to be effective against pathogens causing urinary tract infections [8]. It is also reported for antibacterial effects on *S. aureus*, *Escherichia coli*, and *Proteus vulgaris,* and antifungal activity against *C. albicans* and C. tropicalis [19,20,21]. Furthermore, an in-vitro study on the frankincense oils grown in different regions in Oman showed varying effects against *S. aureus* and *P. aeruginosa*, and a dermatological strain *P. acnes* and a good antifungal effect against *C. albicans* and *M. furfur* [7]. Another study reported antimicrobial activity against five organisms; *S. aureus*, B. cereus, *E. coli*, P. vulgaris, and *C. albicans* [8]. Frankincense is not used alone for the treatment of infections, it is usually combined with myrrh or other antimicrobial agents for antimicrobial effect [22,23]. This study determined the effect of frankincense oil against MRSA and MDR-P. aeruginosa to demonstrate its effectiveness in inhibiting the growth of resistant pathogens. The antibacterial effect of frankincense oil was relatively more against MRSA (Gram-positive) compared to P. aeruginosa (Gram-negative). This is due to the cell wall structure of Gram-negative bacteria that makes it inherently tolerant to the effect of antibacterial agents[24].

The interaction of herbs with conventional antimicrobial drugs is an area of interest and it is being thoroughly investigated to determine both beneficial and adverse reactions that may arise due to the concomitant administration of herbs with drugs [25]. Frankincense oil is a popular herb for the treatment of infections in several countries and it is common for patients to consume it along with antibiotics to obtain ‘extra’ benefits. Despite this, the pharmacological interaction of frankincense oil with antibiotics is not known. In the current study, antibiotics and frankincense oils showed antibacterial effects individually and it is usually expected that a combination of two such active agents produces an additive/synergistic effect [26,27]. Apart from this, some combinations of antibacterial agents such as a combination of bacteriostatic agents with bactericidal drugs induce antagonism of the bacteriostatic effect by bactericidal agents [28]. In the current study, no interaction was observed between the frankincense oil and antibiotics both in-vitro and in-vivo. The reason(s) for this cannot be explained with the present data. However, it can be ruled out that frankincense may alter the pharmacokinetics of concurrently administered antibiotics, as the effect observed was similar in both in-vitro and in-vivo.

The selection of bacterial isolates was based on animal models and pathogens causing opportunistic infections in the lungs [29,30]. MRSA is known to cause pneumonia in hospitalized patients. It is recommended that treatment for MRSA be started if 10–20% of isolates showing antibiotic resistance are MRSA [31]. Imipenem is effective against MRSA and it is one of the drugs used in the treatment of MDR infections in the lungs [32,33]. MDR-*P. aeruginosa* infection is a concern for many physicians as it causes lung function to decline and leads to the emergence of antibiotic resistance in *P. aeruginosa* strains [34]. Though different antibiotics including imipenem/cilastatin have been reported to be effective against MDR-*P. aeruginosa* infections, of late, resistance to imipenem has been reported [35,36]. The MDR strain of *P. aeruginosa* was sensitive to gentamicin, hence, gentamicin was used as a standard drug to study interaction with frankincense oil. The micro-organism selection was also based on two wide groups; the Gram-positive MRSA and Gram-negative *P. aeruginosa* to determine the spectrum of the effect of frankincense oil and its interaction with standard antibiotics in treating infection by two different groups of pathogens. As the organisms were inoculated through the intratracheal route, lung infection will be more severe than infection in other parts of the body. Hence, only the lungs were excised.

The chemical constituents present in frankincense oils differ based on geographical distribution, climatic conditions, and harvesting methods [37]. The chemical composition will vary among the different brands of frankincense oils available in the market due to the above-mentioned factors. Since chemical composition differences will affect the pharmacological activities, the antimicrobial activities of all the brands of frankincense oils cannot be predicted [8]. Hence, chromatographic analysis of essential oils is important. In the current study, 40 different constituents were identified. Different authors have identified different compounds in frankincense oil. Octyl acetate followed by 1-octanol were identified as the main compounds by Baser [38]. Limonene and (*E*)-β-ocimene were found to be the main compounds in *B. sacra* [39]. Camarda et al. [19] also found limonene to be the main component followed by α-pinene as the second component while α-pinene was identified as the main volatile component followed by octyl acetate in Saudi Arabia [40]. In the current study, the oil used contained the maximum amount of α-pinene followed by 1,3,8-p-menthatriene with a total of 40 volatile constituents. Earlier studies have reported the antibacterial activity of some of the constituents revealed by the GC-MS in the current study. The α-pinene and β-pinene have been reported for antibacterial and antifungal effects [41]. Limonene is reported to have an antibacterial effect against *S. aureus* and *P. aeruginosa* and also potentiated the effect of gentamicin in-vitro [42]. Camphene is also reported for antibacterial effect against *S. aureus* and *Enterococcus* species [43], while myrcene is reported for activity against *S. aureus* and food-borne pathogens [44]. Though individual components are known to be effective, it is believed that microbial resistance will not occur against the essential oils because the oil contains several constituents with antimicrobial effects [45]. However, it is also known that the essential oils obtained from different sources may have different effects due to variations in the physicochemical properties of essential oils and the antimicrobial effect obtained in one study cannot be compared with effects obtained with essential oils in another study. This is probably due to the loss of antimicrobial constituents or potentiating compounds during the process of distillation [46].

Our results are different from those reported by several other authors, who showed that frankincense oil has a good antibacterial effect, and a combination of frankincense oil with other antibacterial agents potentiates the effect of the latter. The exact reason(s) for this difference in the effect observed in the current study with earlier reports cannot be explained with the present data. However, the difference in the composition of the oil, the difference in the antibiotic used, and the strain of the micro-organism may have contributed to this difference in the effect. More studies with other micro-organisms and with other antibiotics may provide information to determine the variation in the effect.

For the interaction study, antibiotics were given by the parenteral route (intraperitoneal), while frankincense oil was administered orally. This is a limitation of the study as only the pharmacodynamic interaction between the oil and the standard antibiotics and pharmacokinetic interaction concerning to distribution, metabolism, and excretion of antibiotics was determined without studying the interaction in the intestinal metabolism and absorption. However, we would like to stress here that the pathogens used were resistant to several antibiotics and the best available antibiotic effective against the pathogens was used.

## 4. Materials and Methods

### 4.1. Micro-Organisms

MRSA (ATCC 43300) and MDR-*P. aeruginosa* (ATCC 27853) available at the Department of Clinical Laboratory Sciences was used. The ATCC cultures were sub-cultured in nutrient broth containing glycerin followed by storage at −80 °C. For evaluation of antibacterial effects, the microbes were inoculated in nutrient agar followed by incubation for 24 h at 36 ± 1 °C. The antibiotic susceptibility of the bacterial strains is shown in Table 1 and Table 2. Frankincense oil (Losolin natural oil, Jamal Natural Factory, Medinah, Saudi Arabia) extracted from *Boswellia sacra* oleo gum resin was purchased online through a marketing website.

### 4.2. Animals

Adult Wistar rats weighing between 235 to 245 g and aged between 4.2 to 4.7 months were used. The animals were divided into five groups of six animals each for MRSA and MDR-*P. aeruginosa*. The first group served as control (inoculated with the pathogen), second and third groups of animals received two different doses of frankincense oil at 500 mg/kg and 1000 mg/kg, respectively. The fourth group was treated with antibiotic and the last group was administered a combination of frankincense oil (1000 mg/kg) with the antibiotic. Animals were maintained in the animal house in a separate room to prevent the spread of infection. The Ethical Research Committee of Shaqra University reviewed and approved the experimental procedures on rats (approval number—53/18909).

### 4.3. GC-MS Analysis of Frankincense Oil

A GC-MS 7890A GC system with 5975C VL MSD was used (Agilent Technologies, Santa Clara, CA, USA). Frankincense oil (100 µL) was mixed with 250 µL of water and 750 µL of ethyl acetate. Following this, the upper layer was separated and concentrated. To this, a 50 µL mixture of N,O-Bis (trimethylsilyl) trifluoroacetamide (49.5 µL), and trimethylchlorosilane (0.5 µL) was added followed by the addition of pyridine (10 µL). This was heated for 30 min at 60 °C and the contents were dried using liquid nitrogen before finally dissolving the dried sample (20 mg) in methanol (5 mL) for analysis. After filtration through a 0.22 µm membrane filter, 3 µL was injected through a capillary column (30 m, 0.25 mm, 0.25-micron) with an injector temperature of 270 °C and pressure at 80 kPa. The carrier gas was hydrogen and the total time for analysis was 25 min. The mass spectra obtained were used to identify different compounds by referring to the NIST mass spectral library.

### 4.4. Antibacterial Effect and Determination of MIC and MBC

The culture of MRSA and MDR-*P. aeruginosa* were inoculated into Muller Hinton agar (100 μL; 1.5 × 10^8^ CFU/mL). Wells were made using cork borer and these were loaded with frankincense oil diluted in 10% dimethyl sulfoxide (DMSO) to different concentrations. DMSO helps in the easy diffusion of the oil through the media. Similarly, gentamicin and imipenem were loaded. The inoculated plates were subjected to incubation at a temperature of 36 ± 1 °C for a period of 24 h to measure the zone of inhibition. The antibiotics were selected after an automated antibiotic susceptibility test of MRSA and MRD-*P. aeruginosa* using a Microscan system (McHenry, IL, USA).

For the determination of MIC, Muller Hinton broth was inoculated with 50 μL of liquid cultures of pathogens (0.5 McFarland standard turbidity). Frankincense oil diluted with DMSA to different concentrations was added followed by incubation at a temperature of 36 ± 1 °C for a period of 24 h to determine the MIC. For MBC determination, nutrient agar was inoculated with a loop of culture and incubated at a temperature of 36 ± 1 °C for 24 h period. MBC was the concentration at which no growth was observed.

### 4.5. Interaction Study of Frankincense Oil with Antibiotics against MDR Strains

The synergistic assay of the antibiotics and *B. sacra* oil was determined by the broth dilution method using a checkerboard assay [47]. After serial dilution of the antibiotic, it was added to each well at different concentrations that included sub-inhibitory, inhibitory, and supra-inhibitory concentrations, and the MIC was calculated. Wells with different concentrations of antibiotics and without oil was considered as the MIC for antibiotics. The fractional inhibitory concentration (FIC) index was calculated as follows:FIC index = FIC_frankincense oil_ + FIC_antibiotic_
FIC_frankincense oil_ = MIC_frankincense oil+antibiotic_/MIC_frankincense oil_
FIC_antibiotic_ = MIC_frankincense oil+antibiotic_/MIC_antibiotic_.

The synergistic potential was assessed as if the FIC index is ≤0.5 the combination is synergistic; at more than 0.5 and ≤2 the combination is indifferent and if the FIC index is >2, it is considered antagonistic [42].

### 4.6. Rat Pneumonia Model (In-Vivo)

Albino Wistar rats were anesthetized by intraperitoneal administration of a mixture of ketamine and xylazine (1:10) at a dose of 1 mL/kg [48]. The trachea was exposed surgically and the animals were kept in an inclined position at 60°. The bacterial suspension (MRSA or multi-resistant *P. aeruginosa*) prepared in 1 × phosphate-buffered saline (pH7.4) was administered through the trachea at a dose of 1.2 mL/kg of body weight and the incision was closed. A set of six animals were used for each treatment. Animals were treated with two different doses of frankincense oil at 500 mg/kg/day and 1000 mg/kg/day orally [18], gentamicin (10 mg/kg, i.p) [49], imipenem (120 mg/kg, i.p) [50] alone or in combination, while one group of animals served as control each for MRSA and MDR-*P. aeruginosa*. All the animals were sacrificed after 4 days post-inoculation for assessments of bacterial growth/clearance [51,52]. The lungs were removed and the tissue samples (1 gm) were homogenized for 5 min using phosphate buffer saline (1 mL) under an aseptic technique and the bacterial count was determined after suitable dilutions. Homogenates were serially diluted (up to 10^9^) and plated on nutrient agar. Plates were incubated at 37 °C and the colonies were counted and log colony-forming units (log 10 CFU) were calculated. Lungs were also subjected to histological examination after staining with H & E stain.

### 4.7. Statistical Analysis

Data are presented as mean ± SEM wherever indicated in footnotes. Statistical difference was done using one-way ANOVA followed by Tukey’s test. *p* < 0.05 was considered significant.

## 5. Conclusions

This study aimed to determine the antimicrobial effect of frankincense oil and its interaction with imipenem and gentamicin against MRSA and multidrug-resistant *Pseudomonas aeruginosa*. We demonstrated that frankincense oil showed a modest inhibitory effect against MRSA and MDR-*P. aeruginosa*, the oil did not show a noticeable change in the zone of inhibition when combined with imipenem or gentamicin, indicating a lack of interaction between the oil and the antibiotics. In addition, the FIC determined to evaluate the interaction between the frankincense oil and the antibiotics showed no interaction between the oil and imipenem or gentamicin. Furthermore, there was no interaction between the antibiotics and the frankincense oil in the in-vivo model, and the antibacterial effect was similar to the results obtained in-vitro.

## Figures and Tables

**Figure 1 antibiotics-11-01591-f001:**
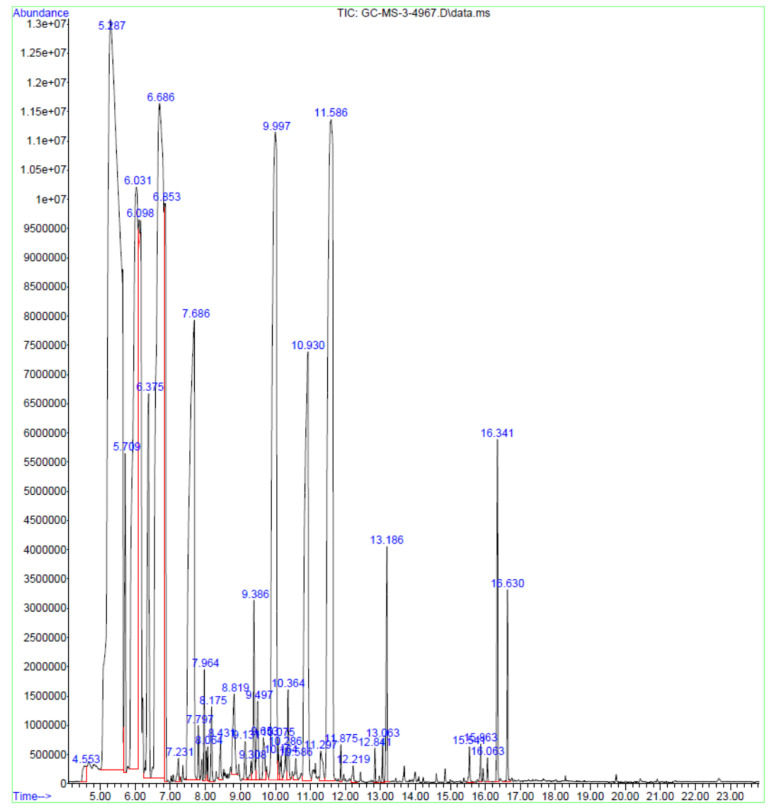
GCMS chromatogram of frankincense oil.

**Figure 2 antibiotics-11-01591-f002:**
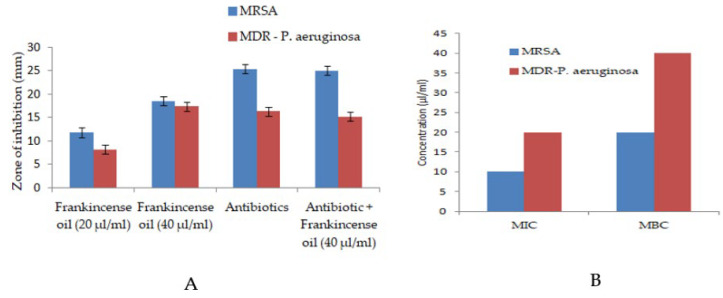
Antibacterial activity of frankincense oil against bacterial pathogens. (**A**) shows zone of inhibition and (**B**) indicates MIC and MBC.

**Figure 3 antibiotics-11-01591-f003:**
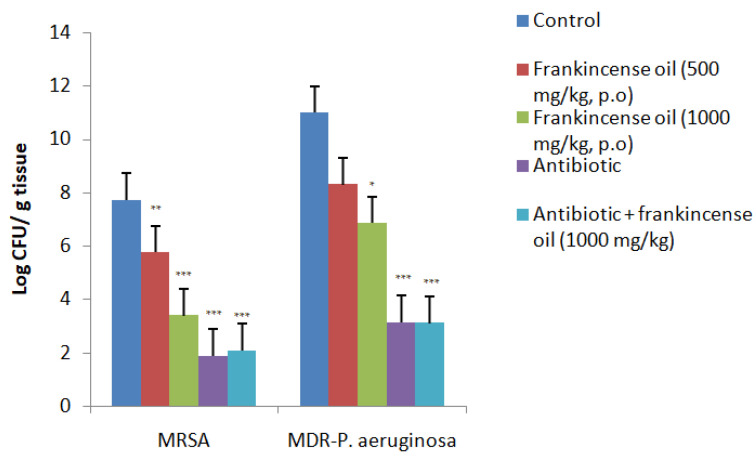
The bacterial load (CFU/g) in the lung tissue after treatment with Frankincense oil. Data shown are mean ± SEM, n = 6, * *p* < 0.05, ** *p* < 0.01, *** *p* < 0.001 when compared to respective controls. There was no significant difference between frankincense oil (500 mg/kg) and MDR-*P. aeruginosa* control. Antibiotic—imipenem (120 mg/kg, i.p) against MRSA, and gentamicin (10 mg/kg, i.p) against MDR-*P. aeruginosa*.

**Figure 4 antibiotics-11-01591-f004:**
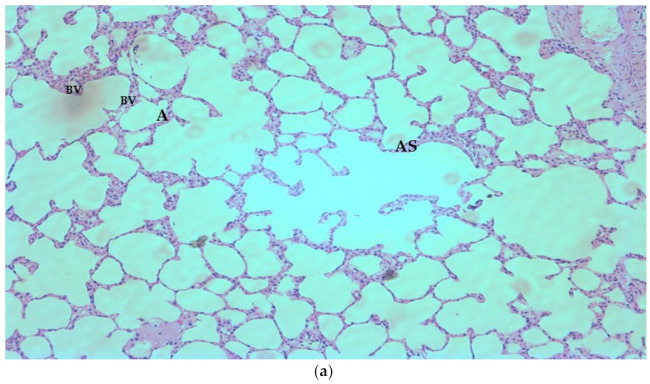
(**a**) Histological examination of lung tissue from normal animals showing alveolar sac (AS), alveolus (A), and blood vessel (BV) (×100). (**b**) Lung tissue histology from *P. aeruginosa* control. A distortion in the pulmonary architecture can be observed with thickened interalveolar septa (black arrows) and mononuclear cellular infiltration (red arrow), interstitial exudates (blue arrow), ruptured interalveolar septa with large irregular emphysematous air spaces (star) are seen (×100). (**c**) Histological examination of lung tissue from frankincense oil-treated group after infection with *P. aeruginosa*. A relatively protected pulmonary tissue showing alveolar sac (AS), blood vessel (BV) with less mononuclear infiltration, reduced thickness of alveolar sacs, and less interalveolar rupture (×100). (**d**) Histological examination of lung tissue from the gentamicin-treated group after infection with *P. aeruginosa*. A relatively protected pulmonary tissue showing alveolar sac (AS), bronchiole (B), and blood vessel (BV) with less mononuclear infiltration, reduced thickness of alveolar sacs, and less interalveolar rupture (×100). The effect was similar to that observed with frankincense oil. (**e**) Histological examination of lung tissue from MRSA control. A distortion in the pulmonary architecture can be observed with thickened interalveolar septa (black arrows) and mononuclear cellular infiltration (red arrow), interstitial exudates (blue arrow), ruptured interalveolar septa with large irregular emphysematous air spaces (star) are seen (×100). (**f**) Histological examination of lung tissue from frankincense oil-treated group after infection with MRSA. A relatively protected pulmonary tissue showing alveolar sac (AS), and bronchiole (B), with less mononuclear infiltration, reduced thickness of alveolar sacs, and less interalveolar rupture (×100). (**g**) Histological examination of lung tissue from imipenem treated group after infection with MRSA. A relatively protected pulmonary tissue showing alveolar sac (AS), and blood vessel (BV), with less mononuclear infiltration, reduced thickness of alveolar sacs, and less interalveolar rupture (×100).

**Table 1 antibiotics-11-01591-t001:** List of constituents detected by GCMS.

Number	Name of the Constituent	RT	Area %
1.	4-Carene	4.553	0.19
2.	α-Pinene	5.287	29.31
3.	Camphene	5.709	1.09
4.	β-Pinene	6.031	9.75
5.	4-Carene	6.098	0.10
6.	α-Phellandrene	6.375	2.29
7.	1,3,8-p-Menthatriene	6.686	15.87
8.	Limonene	6.853	2.61
9.	Cycloheptene, 5-ethylidene-1-methyl	7.231	0.11
10.	1,6-Octadien-3-ol, 3,7-dimethyl-	7.686	7.42
11.	Phenylethyl Alcohol	7.797	0.17
12.	1-Propanone, 1-(5-methyl-2-furanyl)-	7.964	0.25
13.	cis-p-Mentha-2,8-dien-1-ol	8.064	0.08
14.	Bicyclo[3.1.1]hept-3-en-2-ol, 4,6,6-trimethyl	8.175	0.26
15.	Isoborneol	8.431	0.16
16.	Cyclohexane, 1-butenylidene-	8.819	0.56
17.	2-Isopropenyl-5-methylhex-4-enal	9.131	0.17
18.	2-Cyclohexen-1-ol, 2-methyl-5-(1-methylethenyl)-, trans	9.308	0.08
19.	2-Methylbicyclo[4.3.0]non-1(6)-ene	9.386	0.58
20.	1H-Pyrrole-2-carboxaldehyde, 1-methyl-	9.497	0.44
21.	1,2,3,4,4a,5,6,8a-Octahydro-naphthalene	9.653	0.26
22.	Isobornyl acetate	9.997	8.97
23.	Camphene	10.075	0.10
24.	1,6,10,14-Hexadecatetraen-3-ol, 3,7,11,15-tetramethyl-, (E,E)-	10.164	0.11
25.	Bicyclo[3.1.0]hexan-3-ol, 4-methyl-1-(1-methylethyl)-, (1.alpha., 3.beta., 4.beta., 5.alpha.)-	10.286	0.14
26.	Bicyclo[4.1.0]heptan-3-ol, 4,7,7-trimethyl-, (1.alpha., 3.alpha., 4.beta., 6.alpha.)-	10.364	0.34
27.	2,6-Octadien-1-ol, 3,7-dimethyl-acetate, (Z)-	10.586	0.11
28.	2,6-Octadien-1-ol, 3,7-dimethyl-,acetate, (E)-	10.930	4.70
29.	Bicyclo[7.2.0]undec-4-ene, 4,11,11-trimethyl-8-methylene-	11.297	0.24
30.	Caryophyllene	11.586	10.54
31.	alpha.-Caryophyllene	11.875	0.09
32.	Naphthalene, 1,2,3,5,6,7,8,8a-octahydro-1,8a-dimethyl-7-(1-methylethenyl)-, [1R-(1.alpha., 7.beta., 8a.alpha.)]-	12.219	0.07
33.	Caryophyllene oxide	12.841	0.08
34.	Diethyl Phthalate	13.063	0.10
35.	Caryophyllene oxide	13.186	0.76
36.	E,E,E)-3,7,11,15-Tetramethylhexadeca-1,3,6,10,14-pentaene	15.541	0.12
37.	2,6,11,15-Tetramethyl-hexadeca-2,6,8,10,14-pentaene	15.863	0.10
38.	β-Myrcene	16.063	0.07
39.	1,6,10-Dodecatriene, 7,11-dimethyl-3-methylene-, (Z)-	16.341	1.11
40.	2-Propenamide, N-(3,4-dichlorophenyl)-2-methyl-	16.630	0.50

**Table 2 antibiotics-11-01591-t002:** The fractional inhibitory concentration of frankincense oil with different concentrations of antibiotics.

Bacterial Pathogens		Fractional Inhibitory Concentration			Outcome
	MIC Oil (µL/mL)	MIC of Ab (mg/mL)	MIC of Combination	FIC Index	
MRSA ATCC 43300	10	0.002	0.002	1	Indifference
*P. aeruginosa* ATCC 27853	20	0.004	0.004	1	Indifference

FIC index = (MIC_frankincense oil+antibiotic_/MIC_frankincense oil_) + (MIC_frankincense oil+antibiotic_/MIC_antibiotic_). The synergistic potential was assessed as follows; if the FIC index is ≤0.5 the combination is synergistic; at more than 0.5 and ≤2, the combination is indifferent and if the FIC index is >2, it is considered antagonistic.

## Data Availability

Data will be provided on request by writing to the corresponding author.

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
