# Peer review of "In-Vitro and In-Vivo Antibacterial Effects of Frankincense Oil and Its Interaction with Some Antibiotics against Multidrug-Resistant Pathogens"

_antibiotics, 2022, doi:10.3390/antibiotics11111591_

Round 1

Reviewer 1 Report

Though the work presented in the MS is very good however, it is lacking in variety of aspects such as ethnopharmacological info which demands its exploration as an anti-bacterial. 

Some pertinent questions which need to be responded as well as the information to be presented in the MS is as under, 

Q1. Which Frankincense oil (FO) was subjected to GC-MS analysis as you have mentioned that FO from 20 different countries exhibited a varying effect on test panel organisms? Plant information and its geographical location is entirely missing from where the FO is being resourced for the study.  In case it is being commercially procured, what is the authenticity of the product as well as the quality of raw material being used for recovery of the FO?

Q2. The methodology of the FIC index is incorrect.  A checkerboard method is used to develop various combination between the FO and antibiotics.  For this sub-MIC as well as supra-MIC concentrations of antibiotics has to be evaluated (1/16 MIC, 1/8 MIC, 1/4 MIC, 1/2 MIC, MIC, 2 MIC, 4 MIC, 8 MIC).  I am afraid this has not been done and requires immediate attention to be conclusive. 

Q3. How did you arrive to FO dose under in vivo studies (both lower and higher)?  They cannot be arbitrarily chosen but must have a sound basis. Please explain which pharmacological method did you use to arrive to the above doses?

Q4. FO as such cannot be an effective modality for use as an antibacterial agent until it has been used in a modified form or more water-soluble form, then how can you determine its efficacy on animal model? Why did not you derivatize the FO as methyl/ethyl ester for dosing the animals as well as for in vitro evaluation of the FO. The same could have been used for GC-MS analysis giving more reliable analysis for its action.

Q5. Number of animals used in the control set as well as test set missing? How many animals were sacrified? What was the survival rate of MDR-MRSA and MDR- PA animal models in the presence of standard drugs, FO and combination of FO and Standard Drugs? This should also be correlated with respective controls which i find is missing. 

Q6. Why was only Lung excised?  They are also capable of bloodstream infections. 

Q7. Histopathological slides have been given but accompanying data is missing to define the effect of FO individually as well as synergistically.

Q8. Why have you carried out a silylation of FO rather than methylation for GC-MS? 

Apart from the above questions, there are several typographical errors which need to be relooked again.

Author Response

Q1. Which Frankincense oil (FO) was subjected to GC-MS analysis as you have mentioned that FO from 20 different countries exhibited a varying effect on test panel organisms? Plant information and its geographical location is entirely missing from where the FO is being resourced for the study.  In case it is being commercially procured, what is the authenticity of the product as well as the quality of raw material being used for recovery of the FO?

We appreciate the comments of the reviewer and we agree with him that frankincense oil from different countries may have varying effects. In fact, frankincense is given different names such as Boswellia serrata for Asian and African frankincense and Boswellia sacra for frankincense obtained from Oman, a Middle Eastern country. In our case, we selected oil that was prepared from Boswellia sacra as per the manufacturer’s claim. We analyzed the chemical constituents present in the commercially obtained oil by GC-MS so that if a study is repeated with frankincense having the same constituents, a similar effect may be obtained. Furthermore, analysis of the oil revealed the probable constituent(s) that might have contributed to the effect. This is now mentioned in the manuscript (lines 172-178). We have now provided the brand name of the oil in the manuscript (lines 297-298).

Q2. The methodology of the FIC index is incorrect.  A checkerboard method is used to develop various combination between the FO and antibiotics.  For this sub-MIC as well as supra-MIC concentrations of antibiotics has to be evaluated (1/16 MIC, 1/8 MIC, 1/4 MIC, 1/2 MIC, MIC, 2 MIC, 4 MIC, 8 MIC).  I am afraid this has not been done and requires immediate attention to be conclusive. 

We regret the wording used in the methodology. The method followed was a ‘checkerboard assay’ using the broth dilution method. We did not mention the word “checkerboard’ in the manuscript though we had cited a reference for checkerboard assay (number 42 and now changed as number 47 due to addition of other references). This has now been corrected in the manuscript. Now we have corrected in the manuscript that different concentrations of antibiotic were used that included sub-inhibitory, inhibitory, and supra-inhibitory concentrations.

Q3. How did you arrive to FO dose under in vivo studies (both lower and higher)?  They cannot be arbitrarily chosen but must have a sound basis. Please explain which pharmacological method did you use to arrive to the above doses?

Earlier reports indicate frankincense oil is safe and does not induce noticeable toxicity even at higher doses (see reference 1 below). The doses in the study were selected based on earlier reports and a reference has been cited for the same (reference number 44 earlier and now 18 as it is now cited also in the discussion). We would also like to mention that no behavioral difference was observed among the different treated groups indicating a lack of noticeable toxicity and no mortality was seen. Frankincense oil is consumed orally in traditional medicine for the treatment of respiratory disease (see reference 2 below). This is now given in the manuscript (lines 189-193).

  1. Ni, X., Suhail, M.M., Yang, Q. et al.Frankincense essential oil prepared from hydrodistillation of Boswellia sacra gum resins induces human pancreatic cancer cell death in cultures and in a xenograft murine model. BMC Complement Altern Med 12, 253 (2012). https://doi.org/10.1186/1472-6882-12-253
  2. Wakefield, M.E., 2014. Special Treatments: Constitutional Psychospiritual Points. Const. Facial Acupunct. 267–276. https://doi.org/10.1016/B978-0-7020-4947-7.00008-7

Q4. FO as such cannot be an effective modality for use as an antibacterial agent until it has been used in a modified form or more water-soluble form, then how can you determine its efficacy on animal model? Why did not you derivatize the FO as methyl/ethyl ester for dosing the animals as well as for in vitro evaluation of the FO. The same could have been used for GC-MS analysis giving more reliable analysis for its action.

The antibacterial effect of essential oils is a topic of interest and various essential oils have been evaluated and are being used for antimicrobial effects. We have also cited many references related to the antibacterial and synergistic activities of different essential oils in the manuscript. The aim of the study was to determine whether frankincense oil given by oral route prevents the growth of pathogenic bacteria in the lungs. As mentioned above, frankincense oil is consumed orally for the treatment of respiratory diseases. Hence, no derivatization was done and the oil was used as such to mimic its traditional use. This is now mentioned in the manuscript (lines 180-182).

Q5. Number of animals used in the control set as well as test set missing? How many animals were sacrified? What was the survival rate of MDR-MRSA and MDR- PA animal models in the presence of standard drugs, FO and combination of FO and Standard Drugs? This should also be correlated with respective controls which i find is missing. 

A set of six animals were used for each treatment. It is now given in the manuscript (line 357). A detailed grouping of the animals is also given now under the heading ‘Animals’ (lines 302-307). All the animals were sacrificed for the excision of their lungs. It is now given in the manuscript (line 361). Since the experiment was for a short duration of only 4 days, all the animals survived during the experimental period. This is now mentioned in the manuscript (lines 103-104).

Q6. Why was only Lung excised?  They are also capable of bloodstream infections. 

We agree with the reviewer that the pathogens might have entered the bloodstream through lungs. As the organisms were inoculated through the intratracheal route, lung infection will be more severe than infection in other parts of the body. Hence, only the lungs were excised. This is now mentioned in the manuscript (lines 243-245). Furthermore, the procedure used is a standard procedure to induce infection in the lungs in experimental animals and we have cited a reference for the same (reference number 51,52).

Q7. Histopathological slides have been given but accompanying data is missing to define the effect of FO individually as well as synergistically.

The effect of frankincense oil and antibiotics was similar when administered individually and it is now mentioned in the manuscript. There was no synergistic or additive effect and the histological changes observed in the combined group were similar to that observed with the individual treatments. Hence, no photomicrograph from the combination treatment group was given as it will be a repetition of the effect observed with antibiotics or frankincense alone. This is now mentioned in the manuscript (lines 115-117).

Q8. Why have you carried out a silylation of FO rather than methylation for GC-MS? 

The method followed was a standard procedure used in GC-MS. We agree with the reviewer that there are several methods used for derivatization including several methylation procedures. However, we could not get any reference that suggests methylation to be a better derivatization reaction than silylation for the analysis of frankincense oil by GC-MS. Hence, silylation was used. Comparative analysis of essential oils by silylation and different methylation procedures may provide better information about the most suitable methods but it is beyond the scope of this study.

Apart from the above questions, there are several typographical errors which need to be relooked again.

The manuscript has been revised for typographical and grammatical errors.

Reviewer 2 Report

The study describes the "In-vitro and in-vivo antibacterial effects of frankincense oil and its interaction with some antibiotics against multidrug-resistant pathogens". The study did not find any inhibitory effect of frankincense oil against MRSA or MDR-P. aeruginosa, and no interaction was seen between the frankincense oil and imipenem or gentamicin.

I have some minor and major recommendations that need to be addressed carefully.

Abstract:

Line 21-23: Rephrase the conclusion part in the abstract

Introduction:

Line 27: Replace the word "danger" with any other suitable word

Line 32-37: Merge these two paragraphs into one and delete the repetitions. Add more relevant studies. The following two recent studies are better suited here suggested adding here. 

https://doi.org/10.3390/antibiotics11101346

DOI: 10.1186/s13756-018-0303-7

Line 40: Bacterial names should always be in italic. Check throughout the manuscript.

Results:

Line 59-60: what do you mean by highest area? Please rephrase the words

Line 73: "Values are mean±SEM, n=3" What is 3? please explain

Line 74: "gentamicin (10 mcg)" What is mcg? please correct the units 

Line 75-77: The results are not clearly written. Rephrase the sentences

Table 2:

"MIC of Ab+oilFIC Index" Add the units. Please follow the above papers mentioned and get help revising the table

Line 126: "Histological examination of lung tissue from MRSAcontrol" add space between MRSA control

Discussion:

Line 141-144: Revise this paragraph. First, you need to write the importance of Frankincense oil which was previously used as an alternative for MDR. Then start with your results and compare them with other relevant studies.

Line 145-146: "The antibacterial effects of frankincense oil have been reported earlier against a variety of pathogens." Add the missing reference. Which studies have shown the Antibacterial effects of Frankincense oil?

Line 146-150: This sentence is very lengthy and not clear. Please rephrase the sentence and divide the sentence into two parts.

Line 147: "Frankincense oil has been reported" Replace the word reported with "used"

Line 148: "the present determined its antibacterial.........." What does mean by the present? What are you talking about? Do you mean frankincense oil or what?

Line 150-151: Delete this sentence or move it to the last part of the Introduction. No need to show your hypothesis here.

Line 164: "dealt" Change the word dealt

Line 168: "Furthermore, An in-vitro" "An" should be in small

Line 172-173: what does this sentence mean? Are Boswellia and myrrh the same as frankincense?

Line 174-178: Here you saying that frankincense oil is effective against MRSA while previously you mentioned that it has no inhibitory effect against MRSA or any other pathogen. Please explain it. The results should be the same throughout the manuscript.

Line 179: Rephrase the sentence. modern seems not looking  well here

Materials and methods:

Line 263-264: How many rats and how many groups were made? Is there any control group? Please write the details info in this section. 

Line 295: "The synergetic assay" Correct the synergistic word

Line 311: change the word instilled

Line 313: "m/kg/day" Write the units in correct form. Check throughout the manuscript

Line 313: "or imipenem," I think no need to write "or" here

Line 317: "the tissue samples (1 g)" What is g? gm or what

Conclusion:

Line 329-330: Add fullstop after p. aeruginosa.

Line 331: MRSA or MDR-P. aeruginosa" I think or should be replaced with and. Rephrase the words

The introduction part needs more details. please revise and add the importance of AMR and the current trends of MDR. Take help from the following similar studies. DOI: 10.3389/fmicb.2022.1031688/

Author Response

Abstract:

Line 21-23: Rephrase the conclusion part in the abstract

It is done now

Introduction:

Line 27: Replace the word "danger" with any other suitable word

It is changed to ‘threat’

Line 32-37: Merge these two paragraphs into one and delete the repetitions. Add more relevant studies. The following two recent studies are better suited here suggested adding here. 

https://doi.org/10.3390/antibiotics11101346

DOI: 10.1186/s13756-018-0303-7

The two paragraphs have been merged and the references suggested by the reviewer are added now.

Line 40: Bacterial names should always be in italic. Check throughout the manuscript.

Corrections made

Results:

Line 59-60: what do you mean by highest area? Please rephrase the words

The word highest area and least area have been replaced with maximum area under the curve and the minimum area under the curve

Line 73: "Values are mean±SEM, n=3" What is 3? please explain

It is explained now as ‘mean±SEM for three independent trials’

Line 74: "gentamicin (10 mcg)" What is mcg? please correct the units 

Mcg is µg and it is changed now.

Line 75-77: The results are not clearly written. Rephrase the sentences

The sentence is rephrased as the per the reviewer’s suggestion

Table 2:

"MIC of Ab+oilFIC Index" Add the units. Please follow the above papers mentioned and get help revising the table

Corrections have been made by referring to the above papers suggested by the reviewer

Line 126: "Histological examination of lung tissue from MRSAcontrol" add space between MRSA control

Correction made

Discussion:

Line 141-144: Revise this paragraph. First, you need to write the importance of Frankincense oil which was previously used as an alternative for MDR. Then start with your results and compare them with other relevant studies.

Correction made as per the reviewer’s suggestion

Line 145-146: "The antibacterial effects of frankincense oil have been reported earlier against a variety of pathogens." Add the missing reference. Which studies have shown the Antibacterial effects of Frankincense oil?

References given now.

Line 146-150: This sentence is very lengthy and not clear. Please rephrase the sentence and divide the sentence into two parts.

Sentence is shortened now.

Line 147: "Frankincense oil has been reported" Replace the word reported with "used"

It is done now.

Line 148: "the present determined its antibacterial.........." What does mean by the present? What are you talking about? Do you mean frankincense oil or what?

It is changed now as ‘present study’

Line 150-151: Delete this sentence or move it to the last part of the Introduction. No need to show your hypothesis here.

The sentence is moved to the introduction as per the suggestion of the reviewer.

Line 164: "dealt" Change the word dealt

Sentence is modified now

Line 168: "Furthermore, An in-vitro" "An" should be in small

Corrected

Line 172-173: what does this sentence mean? Are Boswellia and myrrh the same as frankincense?

Boswellia is frankincense. The word Boswellia is now replaced with frankincense.

Line 174-178: Here you saying that frankincense oil is effective against MRSA while previously you mentioned that it has no inhibitory effect against MRSA or any other pathogen. Please explain it. The results should be the same throughout the manuscript.

We have mentioned in the manuscript that frankincense does not have a ‘potent’ inhibitory effect against MRSA or MDR-P. aeruginosa. This sentence was written to explain that the antibacterial effect of frankincense was ‘relatively’ more against MRSA compared to MDR-P. aeruginosa. This correction is made now in the manuscript.

Line 179: Rephrase the sentence. modern seems not looking  well here

The word ‘modern’ is now replaced with ‘conventional antimicrobial’

Materials and methods:

Line 263-264: How many rats and how many groups were made? Is there any control group? Please write the details info in this section. 

A detailed explanation is given now as suggested by the reviewer.

Line 295: "The synergetic assay" Correct the synergistic word

Corrections made

Line 311: change the word instilled

The word ‘instilled’ is changed to ‘administered’

Line 313: "m/kg/day" Write the units in correct form. Check throughout the manuscript

Correction made and checked

Line 313: "or imipenem," I think no need to write "or" here

Correction made

Line 317: "the tissue samples (1 g)" What is g? gm or what

Correction made

Conclusion:

Line 329-330: Add fullstop after p. aeruginosa.

Correction made

Line 331: MRSA or MDR-P. aeruginosa" I think or should be replaced with and. Rephrase the words

Correction made

The introduction part needs more details. please revise and add the importance of AMR and the current trends of MDR. Take help from the following similar studies. DOI: 10.3389/fmicb.2022.1031688/

The reference suggested by the reviewer has now been cited as a support for AMR.

Round 2

Reviewer 2 Report

The authors have addressed all of my concerns and suggestions. However, after reviewing again I found some minor errors which need to be addressed before publication.

1. Line 77-79: "Values are mean±SEM for three independent trials, trials Antibiotic imipenem (4 µg/ml) against MRSA, and gentamicin (10 µg) against MDR-P. aeruginosa" Rephrase this sentence also remove the repeated words

2. Line 84: "2.3. Evaluation of the synergistic effect of antibiotics and oil on MDR strains," write the full name "Frankincense oil". not just oil.

Better to rewrite the heading as "Synergistic combine effect of Frankincense oil with other antibiotics against MDR .........." Rephrase it accordingly which fits better.

Author Response

The authors have addressed all of my concerns and suggestions. However, after reviewing again I found some minor errors which need to be addressed before publication.

  1. Line 77-79: "Values are mean±SEM for three independent trials, trials Antibiotic imipenem (4 µg/ml) against MRSA, and gentamicin (10 µg) against MDR-P. aeruginosa" Rephrase this sentence also remove the repeated words.

The repeated word is deleted now. The sentence has been rephrased as “Values are mean±SEM for three independent trials. Antibiotic refers to imipenen (4 µg/ml) and gentamicin (10 µg/ml) against MRSA and MDR-P. aeruginosa respectively”

  1. Line 84: "2.3. Evaluation of the synergistic effect of antibiotics and oil on MDR strains," write the full name "Frankincense oil". not just oil.

Better to rewrite the heading as "Synergistic combine effect of Frankincense oil with other antibiotics against MDR .........." Rephrase it accordingly which fits better.

The word ‘frankincense’ has been added now. The title has been modified as “Interaction study of frankincense oil with antibiotics against MDR strains” as not synergistic effect was observed. This correction is made in both the results and methodology sections.